# Distinct transcriptomic profiles in children prior to the appearance of type 1 diabetes-linked islet autoantibodies and following enterovirus infection

Jake Lin[1,2,3,4,19], Elaheh Moradi[1,5,19], Karoliina Salenius [1,19], Suvi Lehtipuro[1], Tomi Häkkinen[1], Jutta E. Laiho[6], Sami Oikarinen[6], Sofia Randelin[1], Hemang M. Parikh [7], Jeffrey P. Krischer[7], Jorma Toppari[8,9], Åke Lernmark [10], Joseph F. Petrosino[11], Nadim J. Ajami [11,12], Jin-Xiong She[13], William A. Hagopian [14,15], Marian J. Rewers [16], Richard E. Lloyd[11], Kirsi J. Rautajoki [1] ✉, Heikki Hyöty [6,17] ✉, Matti Nykter [1,18] ✉ & the TEDDY Study Group*

Although the genetic basis and pathogenesis of type 1 diabetes have been studied extensively, how host responses to environmental factors might contribute to autoantibody development remains largely unknown. Here, we use longitudinal blood transcriptome sequencing data to characterize host responses in children within 12 months prior to the appearance of type 1 diabetes-linked islet autoantibodies, as well as matched control children. We report that children who present with insulin-specific autoantibodies first have distinct transcriptional profiles from those who develop GADA autoantibodies first. In particular, gene dosage-driven expression of *GSTM1* is associated with GADA autoantibody positivity. Moreover, compared with controls, we observe increased monocyte and decreased B cell proportions 9-12 months prior to autoantibody positivity, especially in children who developed antibodies against insulin first. Lastly, we show that control children present transcriptional signatures consistent with robust immune responses to enterovirus infection, whereas children who later developed islet autoimmunity do not. These findings highlight distinct immune-related transcriptomic differences between case and control children prior to case progression to islet auto-immunity and uncover deficient antiviral response in children who later develop islet autoimmunity.

Type 1 diabetes (T1D) is a chronic disease characterized by immune-mediated loss of functional pancreatic islet beta cells. It has been estimated that around 40–50% of the risk of the disease arises from genetics, while half of this genetic risk links to class I and class II human leukocyte antigen (HLA) genes. Environmental factors also play an important role in the pathogenesis and contribute to the rapid increase in the disease incidence that has occurred during the past decades. Among them, dietary factors and virus infections have been

A full list of affiliations appears at the end of the paper. *A list of authors and their affiliations appears at the end of the paper. ✉e-mail: kirsi.rautajoki@tuni.fi; heikki.hyoty@tuni.fi; matti.nykter@tuni.fi

studied widely, but the exact triggers and mechanisms leading to islet autoimmunity and the subsequent beta cell destruction have still remained without final confirmation. The appearance of islet auto-antibodies (IAbs), directed against beta cell antigens insulin (IAA), 65-kDa isoform of glutamic acid decarboxylase (GADA), insulinoma antigen 2 (IA2A), or zinc transporter 8 (ZnT8A), is the earliest sign of immune-mediated pathogenesis revealed by indirect immuno-fluorescence already in 1974[1,2]. The IAbs occur usually long before clinical T1D is diagnosed and often already during the first years of life. The mechanisms of the immunological process leading to the appearance of first IAb (IAb seroconversion) and eventually to the clinical disease are complex, and it is widely accepted that T1D pathogenesis has a high degree of disease heterogeneity[3].

HLA-DR3-DQ2 and HLA-DR4-DQ8 extended haplotypes account for the highest genetic risk for T1D. While the appearance of IAA peaks sharply before one year of age and shows association with DR4-DQ8 haplotype, the appearance of GADA is more commonly associated with HLA-DR3-DQ2 and occurs usually later, starting to increase during the second year of life and staying at relatively constant rate thereafter[4–7]. Other known genetic markers of the appearance of IAbs include e.g. *PTPN22* and *INS*[8,9]. In addition, heightened inflammation and aberrant lipid pathways have been reported to increase risk[10–12].

Recently, the concept of different disease subtypes has emerged[4,13–15]. Currently, at the initiation of the autoimmune process, two major pathways with potentially different etiopathogenesis have been recognized with either IAA or GADA as the first appearing IAb. Interestingly, the IAA-first subtype, but not GADA-first, has been shown to be associated with coxsackievirus B1 (CVB1) infections[16], belonging to enterovirus B (EV-B) species. The recent discovery of these different disease subtypes opens up new opportunities to understand the heterogeneous nature of T1D pathogenesis.

The Environmental Determinants of Diabetes in the Young (TEDDY) study is among the largest prospective birth cohort studies[9,17] of newborns, evaluating the role of adverse genetic risk together with environmental factors in the pathogenesis of T1D. TEDDY established a nested case-control (NCC) design where study subjects have been extensively characterized with multiple omics technologies from longitudinal samples, collected during quarterly visits, and metagenomic sequencing from monthly stool and plasma samples until islet autoimmunity or T1D onset. The role of virus infections in the initiation of islet autoimmunity has been actively studied in TEDDY, and it was recently reported that human adenovirus (HAdV) and enterovirus (EV) infections, and particularly prolonged course of EV-B infections, were associated with increased risk of islet autoimmunity[18–20].

Earlier studies have characterized the interactions of different pathways and transcriptional networks prior to the appearance of IA. Previous blood transcriptomics analyses carried out in children with HLA-conferred T1D susceptibility have shown that innate immunity functions, such as the type 1 interferon (IFN) response signatures are activated prior to IAb seroconversion[21–23]. Notably and recently reported in the TEDDY study, age-related genetic network signatures were identified across islet autoimmunity being common for both GADA-first and IAA-first subtypes[24]. The investigators further identified these genetic modular signatures to contain strong enrichments in B- and NK- cell associated transcription profiles and subsequently, the NK-cell progression signature was validated using a cohort from the Type 1 Diabetes Prediction and Prevention (DIPP) study[25].

With the whole blood transcriptome sequencing data from TEDDY longitudinal design, we aim to elucidate temporal changes in gene expression and immune cell proportions in different autoantibody patterns and in EV infections prior to IAb seroconversion.

## Results

### Characteristics and harmonization of the TEDDY nested case-control islet autoimmunity cohort

Whole blood samples collected from 418 case-control pairs (1:1) of children included in the TEDDY Nested Case-Control islet auto-immunity cohort 1 (NCC1)[9,17] were used for whole transcriptome sequencing. Case children included children who had developed at least one biochemical IAb during the prospective observation (positivity was confirmed by two laboratories in at least two consecutive samples). One control child who had remained constantly IAb negative and was matched for biological sex, clinical TEDDY site and family history with T1D was selected for each case child (Supplementary Data 1a Demographics[18,19]). Rigorous quality control was performed at all steps of the data generation (see Methods). For our analysis, transcriptome data was harmonized with next-generation sequencing (NGS) virome data, resulting in data from 1693 samples within 312 NCC1 islet autoimmunity pairs (mean age of IAb seroconversion 727 days), including 140 IAA-first (mean age of IAb seroconversion 636.1 days; IQR 443–844), 105 GADA-first (mean age of IAb seroconversion 921.8 days; IQR 658–1258), and 67 other (at least two of IA2A, GADA or IAA at the same first autoantibody positive sample, mean age of IAb seroconversion 799.4 days; IQR 550–961), available for analysis (see Methods). To understand the temporal expression patterns prior to IAb seroconversion, we binned the NCC1 samples by the sample due month identifier to indicate the matching time points in the case and matched control children in relation to three month intervals ranging from time of IAb seroconversion (first autoantibody positive sample) to 12 months prior (Fig. 1a, Supplementary Data 1b, c).

### Prior gene expression patterns are distinct in children with IAA or GADA seroconversion

Statistical testing was performed at each time interval (see Methods) to identify an initial set of candidate genes with differential expression between the cases and controls (adjusted p-value<0.05, absolute log fold change (|LFC|)>0.5). In the full islet autoimmunity cohort, this resulted in 18 differentially expressed genes across time points (Fig. 1b, Supplementary Data 2a). One of these genes (*RPS26*) mapped to loci from a prior T1D genome wide association study (GWAS)[26]. To address if children with different IAb profiles have similar trajectories towards seroconversion, we performed the same analysis for sample groups with GADA or IAA detected as the first autoantibody. For the GADA-first group we identified 181 genes, of which 4 genes (*IKZF3*, *CDKN1C*, *RPS26*, *IL7B*) mapped to T1D GWAS loci (Fig. 1b, Supplementary Data 2b). In the IAA-first group we identified 36 genes of which no genes mapped to T1D GWAS loci (Supplementary Data 2c, Fig. 1b). This analysis revealed distinct trajectories towards seroconversion in these two groups (Fig. 1b) - both the sets of differentially expressed genes as well as the temporal dynamics of the differential expression were characteristic for each group (Fig. 1b, c). In the IAA-first cohort, most differentially expressed genes were detected at 9 months before the IAA seroconversion, while in the GADA-first cohort, most genes were detected at 12–18 months prior to GADA seroconversion. Adjusted *p*-values and LFCs for selected genes and all timepoints for the full islet autoimmunity NCC1, GADA-first, and IAA-first cohorts are listed in Supplementary Data 2a–c and expression profiles visualized in Supplementary Figs. 1–3. Pathway analysis with Enrichr gene set enrichment analysis[27] using all the differentially expressed genes (Supplementary Data 2) revealed enrichment in complement activation, in addition to intracellular processes such as cell cycle. Concordantly, enrichment of the complement cascade and its regulation was confirmed in plasma proteome data[28] from the same samples (see Methods, Supplementary Data 3).

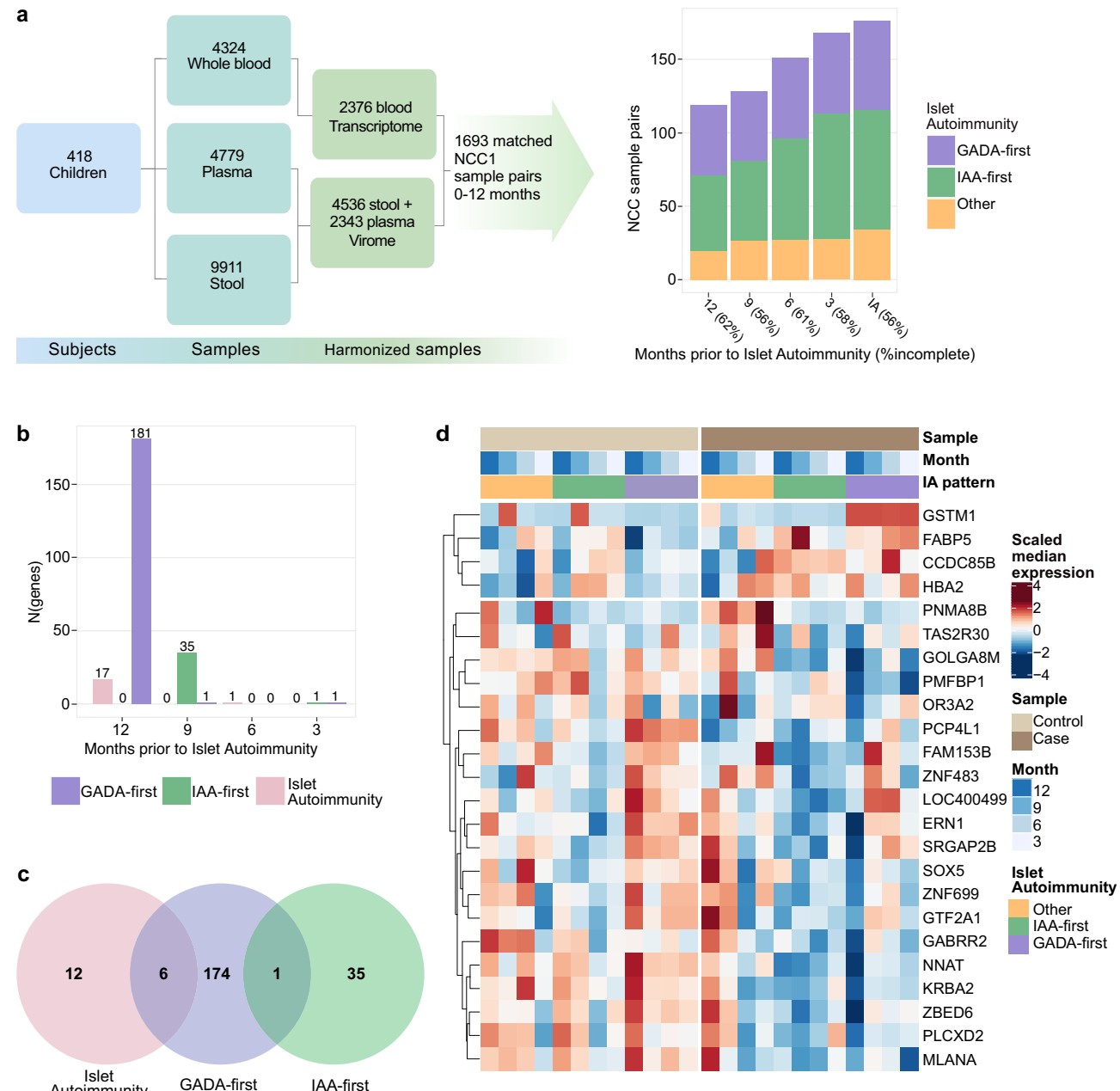

**Fig. 1 | Temporal gene expression patterns differ in IAA-first and GADA-first islet autoimmunity seroconversion types. a** TEDDY Nested Case-Control (NCC1) islet autoimmunity cohort and subsequent virome and sample harmonization yielding 1693 matched samples in 312 case-control pairs. These samples are further mapped as GADA-first (105 pairs in which the case child developed GADA as the first appearing IAb), IAA-first (140 pairs with IAA as the first appearing IAb) and Other (67 pairs with GADA and IAA seroconverted at the same time or IA2A detected first). **b** On each time point prior to seroconversion, statistical testing using DESeq2 is applied to compare the gene expression in case and matched control. The bar chart indicates the selected number of genes passing the threshold of adjusted *p*-value <0.05 and |LFC|>0.5. The statistics for the genes significant at each time point are listed in Supplementary Data 2. **c** The Venn diagram shows distinct gene sets for full islet autoimmunity, IAA-first, and GADA-first cohorts. **d** Differentially expressed genes with consistent temporal pattern selected from the full islet autoimmunity, GADA-first and IAA-first cohorts. For visualization expression of each gene is scaled to have zero mean and unit variance.

## Temporal filtering identifies subset of disease relevant genes as potential biomarkers

We narrowed the analysis further by selecting genes that in addition to statistical significance in a single time point have a temporal expression alteration only in cases between two consecutive time points (see Methods). When these additional criteria were applied, 2 genes in the full islet autoimmunity cohort, 14 genes in GADA-first and 1 gene (*FABP5*) in IAA-first remained (Supplementary Figs. 4–7, Supplementary Data 4). We applied conditional logistic regression (see Methods) in the NCC1 setting to test the association of the expression of each of

the selected genes with temporal change to islet autoimmunity using HLA genotype of the subject as a covariate in the model (see Methods). All of the selected genes remained statistically significant in one or more time points while 4 of the 17 genes were significant for multiple time points (Fig. 2a, Supplementary Data 5).

Many of these genes have previously been linked to T1D relevant biological processes. *GSTM1* is an enzyme that detoxifies electrophilic compounds, including prostaglandins[29,30]. Frequent deletions in the gene (frequency 40–60% in European ancestry populations) generate a null, *GSTM1* inactive, genotype[31]. Previous studies have associated

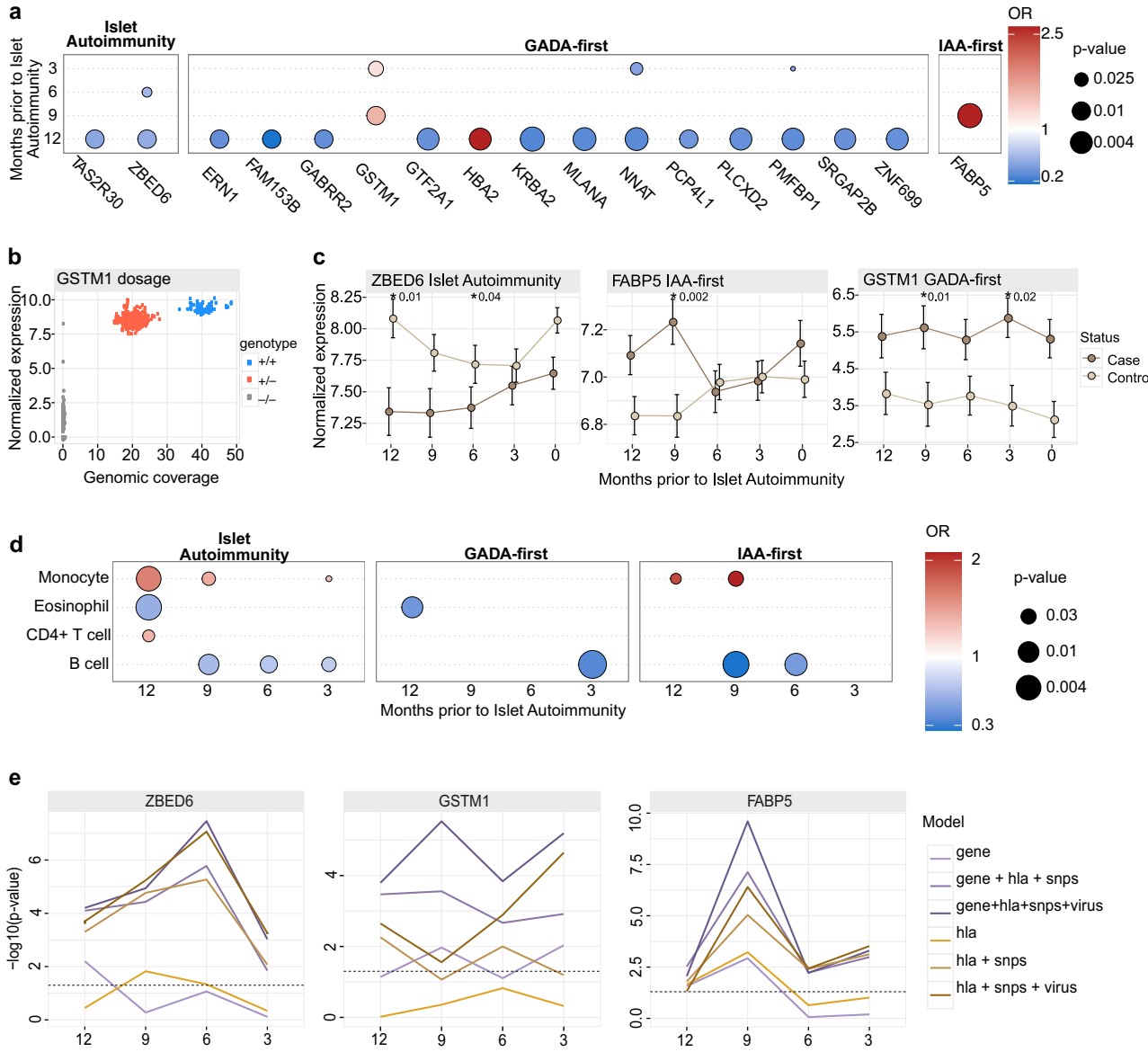

**Fig. 2 | Distinct temporal patterns across selected islet autoimmunity associated genes and immune cell types. a** Conditional logistic regression analysis for differentially expressed genes between case and control children in different time points prior to IAb seroconversion. Associated *p*-values are indicated by the dot size and odds ratios (OR) by shades of red (>1, positively associated to islet autoimmunity NCC1, implying a higher risk for higher gene expression) and blue (<1, negatively associated to islet autoimmunity NCC1, which can be inferred as the higher gene expression being protective). *P*-values are unadjusted due to the nested case-control setting. **b** *GSTM1* gene dosage effect in the full islet autoimmunity NCC1 cohort. Genotype is indicated with color: +/+ is for diploid, +/− for hemizygous deletion, and −/− for homozygous deletion. **c** Line plots covering 0-12 months prior to seroconversion showing the mean temporal differences in islet autoimmunity NCC1 cases and controls as illustrated by representative genes, selected based on known biological function, *ZBED6* (full islet autoimmunity NCC1), *GSTM1* (GADA-first) and *FABP5* (IAA-first), with error bar showing the standard error of the mean (±sem) and highlighting the significant *p*-values derived from conditional logistic regression analysis (Supplementary Data 4). Temporal expression patterns for other genes are shown in Supplementary Figs. 5–7 and their detailed statistics, including log fold change and adjusted p-value are listed in the Supplementary Data 4. **d** Conditional logistic regression analysis identifies distinct immune cell type profiles for the full islet autoimmunity NCC1, GADA-first, and IAA-first seroconversion sets. IAA-first is shown to have more aberrant cell type profiles. The dot size and color have the same meaning as in **a**. **e** Using likelihood ratio test and as shown with *ZBED6* (full islet autoimmunity NCC1), *GSTM1* (GADA-first) and *FABP5* (IAA-first) models, discrimination performances were consistently improved with the inclusion of gene, hla, SNPs and virus. Other putative temporal markers are shown in Supplementary Data 8 and Supplementary Fig. 12.

*GSTM1* null phenotype with a decreased risk of T1D[32], while our results further highlight the potential role of high *GSTM1* expression as a risk factor for GADA-first cases (higher expression associated with a higher risk to the GADA-first group). To test if observed *GSTM1* gene expression is genetically driven, we quantified the genotype of the *GSTM1* in each sample with available whole genome sequencing data (see Methods). Analysis confirmed *GSTM1* expression regulation through gene dosage effect (Fig. 2b) as well as risk association for

GADA-first cases at the genotype level (*p* = 0.0045, Fisher's exact test; Supplementary Fig. 8).

Other relevant genes include *ZBED6* (downregulated in cases in the full islet autoimmunity at 6 and 12 months) and *FABP5* (upregulated in cases in the IAA-first group). Notably, *ZBED6* has been demonstrated to repress *IGF2*[33] and has a positive role in the regulation of pancreatic beta cell survival[34]. *FABP5*, a fatty acid binding protein, is known to modulate inflammation and associates also with monocyte activation

in early T1D[35–38]. The temporal differences of *GSTM1*, *FABP5* and *ZBED6* between islet autoimmunity cases and controls from months 0 to 12 are illustrated in Fig. 2c.

## Proportions of leukocyte subpopulations are altered prior to islet autoimmunity

We further extended the analysis from the gene expression levels to the analysis of relative proportions of different leukocyte subpopulations. After validating the performance with an external dataset (see methods and Supplementary Fig. 9) we applied gene expression deconvolution analysis to infer regression coefficients for leukocyte subpopulations in each sample (Supplementary Fig. 10). Validation data demonstrates that these coefficients are related to the abundance of cell types in the sample (Supplementary Fig. 9). Pairwise relationships between inferred cell type proportions are shown in Supplementary Fig. 11. Statistical analysis using a conditional logistic regression model (see Methods) uncovered different leukocyte profiles in case children compared to control children (Fig. 2d), and distinct differences in the full islet autoimmunity NCC1, GADA-first, and IAA-first seroconversion cohorts (*p*-value < 0.01, Supplementary Data 6). As shown in Fig. 2d, decreased B lymphocyte (B cell) proportion was associated with increased risk of IAb seroconversion in the full islet autoimmunity cohort across the time course. Decreased B cell proportion was also observed in GADA-first at 3 months and for IAA-first children at 6 and 9 months before IAb seroconversion (Fig. 2d, Supplementary Fig. 10, Supplementary Data 6). In addition, the case children in the IAA-first and the full islet autoimmunity cohorts had an elevated monocyte component at 9 and 12 months prior to islet autoimmunity (Fig. 2d). This is known to be associated with chronic inflammation[39,40]. Interestingly, eosinophil cell proportions were found to be protective in the full islet autoimmunity and GADA-first patterns in a single time point at month 12 (full islet autoimmunity 0.59 (0.42–0.84), GADA-first 0.45 (0.25–0.82)). IAA-first cases have more aberrant leukocyte profiles than GADA-first cases, in relation to their matched controls, across the cell types responsible for both innate and adaptive immune responses.

## Analysis of stool and plasma virome confirms association between enterovirus infections and islet autoimmunity

To enable integrative analysis with the virus infections, we first reanalyzed the NGS virome data from the matching stool[18] and plasma samples and refined the specificity of the virome results by performing viral capsid based virus genotyping focusing on EV and human adenovirus (HAdV) species[41] (see Methods). Consistent with previously reported results[18,19], we found a significant risk association of CVB group EV infections (HLA adjusted) with the appearance of the islet autoimmunity NCC1 (OR 1.80 (1.33–2.43)), IAA-first (OR 1.79 (1.07–2.99)) and GADA-first (OR 1.71 (1.14–2.57)). For all EV infections and consistent with prior TEDDY results, we detected an OR (HLA adjusted) of 1.31 (1.15–1.73) for full islet autoimmunity NCC1, OR of 1.49 (1.03–2.14) for IAA-first and OR of 1.30 (0.98–1.71) for GADA-first. Interestingly, after excluding CVB infections from all EV infections, we found insignificant ORs (HLA adjusted) of 1.15 (0.84–1.58) for full islet autoimmunity NCC1, 1.14 (0.67–1.93) for IAA-first, and 0.92 (0.58–1.44) for GADA-first. Additional virome model statistical details are shown in Supplementary Data 7.

## Expression markers enable improved stratification of islet autoimmunity outcomes into IAA-first and GADA-first autoantibody patterns

To evaluate if host transcriptome can contribute to patient stratification on top of the previously established markers[7], we constructed logistic regression models using above mentioned genetic markers and viral infections, and the transcriptomic markers identified in our discovery analysis. We further quantified the model performance using likelihood ratio test *p*-value and demonstrated that the prediction is improved with inclusion of transcriptomic markers and viruses, consistently across months (3, 6, 9 and 12) prior to IAbs conversion. Along with *HLA*, *PTPN22* and *INS* SNP markers reported in a prior TEDDY study[7] we found improvements in model prediction, confirmed by comparing likelihood ratio tests on all relevant timepoints, after incorporation of EV and HAdV infections with the selected genes across islet autoimmunity NCC1 (2 genes), GADA-first (14 genes) and IAA-first (1 gene) conversions (Fig. 2e, Supplementary Data 8, Supplementary Fig. 12). This demonstrates that incorporation of these transcriptomic markers have power to improve the stratification of children beyond the established genetic and environmental markers.

## A robust host immune response to enterovirus infection is observed in children who do not develop islet autoimmunity

The possible contribution of virus-induced host responses to the initiation of islet autoimmunity is not known. To address this, we extended our analysis to study transcriptome differences between case and control children in the full islet autoimmunity cohort from sample pairs collected before and after infections by different EV and HAdV types. To identify virus induced genes, we performed integration with virome data and tested for differentially expressed genes at the initial virus infections (first EV or HAdV infection in the child), diagnosed by detecting virus genome sequences in stool or plasma, with the closest prior infection-free transcriptome of the same child (see Methods). In this setting, for EV, our data set yielded 44 cases (including 9 EV detections from plasma) and 13 control children (including 5 EV detections from plasma) with available transcriptome data before and after EV infection (EV+). Subject and infection details, including HLA, infection age and selected EV-B strain infection states, are listed in Supplementary Data 9. For HAdV, our data yielded 88 islet autoimmunity cases and 83 control sample pairs with available transcriptome data before and after HAdV infection (HAdV+).

By statistical testing (DESeq2 adjusted *p*-value < 0.05,|LFC|>1) we detected 37 differentially expressed enterovirus induced genes in the control children and none in case children (Fig. 3a, b, Supplementary Data 10 (sorted by LFC)). Notably, all the differentially expressed genes were upregulated, including the chemokine ligand *CXCL10* gene which has been reported to be upregulated during EV infections in pancreatic islets[42–44]. Supportive results were obtained for the genes with matching proteome plasma data available (C2, SERPING1; see Methods) (Supplementary Fig. 13). Notably, 70% (26/37) of the genes upregulated upon EV infections in control children belonged to the innate immunity system pathway[45,46]. In addition, based on DAVID gene set enrichment analysis test[47] these upregulated genes are involved in positive regulation of immune response and antiviral response (enriched pathways (adjusted *p*-value < 0.05, minimum of 3 genes), Supplementary Data 11a). In parallel analysis using HAdV infections, induction of only a single differentially expressed gene with |LFC|>1, *PI3*, a known pathogen inhibitor[48,49] was detected from 83 control samples and no genes with |LFC|>1 from 88 case pairs (Supplementary Data 12). We also refined the HAdV analysis for HAdV-F, reported as a T1D islet autoimmunity risk[18], and detected no genes with |LFC|>1 (Supplementary Data 12).

We further confirmed our observation of EV response by excluding the |LFC|>1 criteria from the analysis, leading to an extended set of 483 genes (of which 368 were upregulated) in control children and 28 genes (of which 16 were upregulated) in cases (adjusted *p*-value < 0.05, Supplementary Fig. 14, Supplementary Data 10a cases, 10b controls). Processes related to positive regulation of immune response, interferon signaling, and viral response were enriched in controls, whereas enrichment to certain developmental processes and chemokine ligand interaction was detected in cases (Supplementary Data 11b, c). In contrast, for HAdV, we detected 2680 genes from the control children (Supplementary Data 11b), 1541 of which were

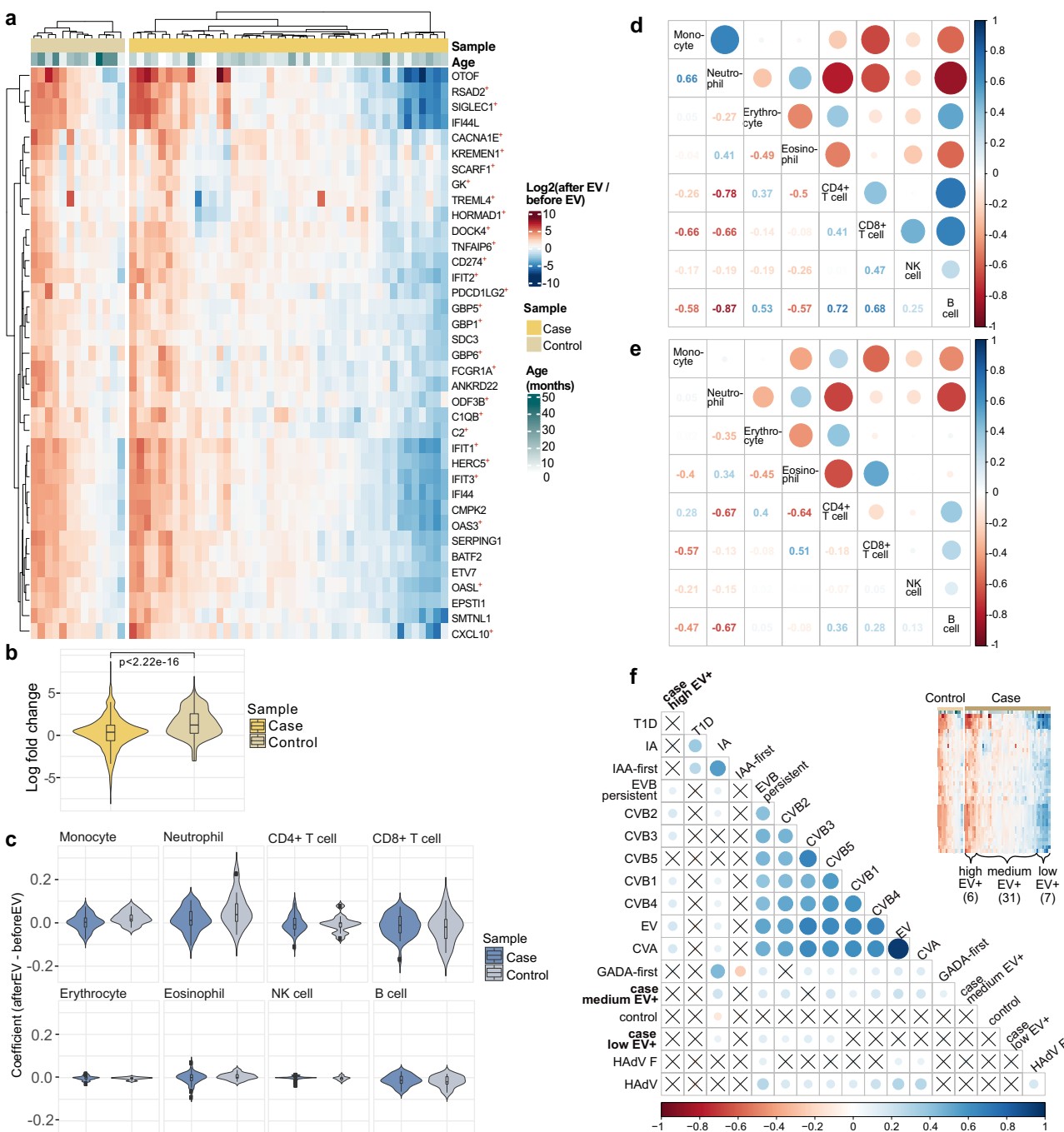

**Fig. 3 | Multi-omics analysis reveals differences in antiviral response between case and control children prior to islet autoimmunity. a** 37 differentially expressed (DESeq2 adjusted $p < 0.05$, |LFC|>1) genes (DEGs) were detected in control children when comparing between samples prior and after enterovirus infection (EV+). All detected genes were upregulated after infection. Among islet autoimmunity NCC1 cases no DEGs were detected. 26 of the genes detected from control, indicated with +, are linked to the innate immunity (Welsh paired T-test, $p$-value 7.01e−06), suggesting a clear innate immune response in controls. Heatmap visualizes the individual LFCs of gene expression. **b** Distribution of LFCs for the genes in Fig. 3a (two-sided Wilcoxon rank sum test). **c** The EV infections related differences in estimated cell type proportions for cases and controls. Neutrophil proportions increase upon EV infection. In controls, monocyte proportions tend to increase while CD8+ T cell proportions decrease upon EV infection. Conditional

logistic regression results in Supplementary Data 14. Correlation map between different immune cell types in controls (**d**, Supplementary Data 13a) and cases (**e**, Supplementary Data 13b) based on the samples taken before and after EV infection. The rows and columns correspond to the cell types. In the upper triangle, circle color and size indicate the Pearson correlation coefficient and the lower triangle shows the exact values. **f** Correlation map of seroconverted cases partitioned by strength of immune response (high, medium, low) upon EV infections and host inflammation episodes. The low responders, partitioned on the right, are found to be correlated with coxsackievirus B (CVB) but not coxsackievirus A (CVA) and human adenovirus F (HAdV F). Circle color and size indicate the associated Pearson correlation coefficient and coefficients with $p$-value > 0.05 are crossed over. For boxes in **b**&**c**: center lines, median; box limits, upper and lower quartiles; whiskers, values within $1.5 \times IQR$ of the top and bottom quartiles.

upregulated. Only 150 differentially expressed genes (Supplementary Data 12) were detected from the HAdV+ cases, of which 44 were upregulated. Thus, unlike with EV+ data, enrichment for upregulation was not observed (*p*-value 0.25).

### Altered leukocyte proportions upon enterovirus infection support robust immune response in control children

Interestingly, neutrophil-related and cytokine-induced responses were enriched during EV infection in both case and control children (Fig. 3c and Supplementary Data 11). This is supported by the trend of increased proportion of neutrophils in the blood upon infection (Fig. 3c) and their positive correlation with the upregulated genes especially among controls (Supplementary Fig. 15). Viral infection related neutrophil profiles negatively correlated with CD4+ T cell (−0.67 in cases, −0.78 in controls) and B cell (−0.67 in case, −0.87 in controls) profiles in both cases and controls, but the negative correlation between neutrophils and CD8+ T cells was detected only in controls (−0.66 in controls, −0.13 in cases) (Fig. 3d, e). Neutrophils and CD8+ T cells were also negatively correlated both in cases and controls upon HAdV (−0.48 in controls, −0.45 in cases) (Supplementary Fig. 16a, b), thus differentiating EV+ case patterns from the others. Similarly, CD4+ and CD8+ T cells were positively correlated only in EV+ controls (correlation coefficient 0.41) (Fig. 3d, Supplementary Fig. 16a) but not in EV+ cases or after HAdV infection (Fig. 3e, Supplementary Fig. 16b). Furthermore, an increased proportion of monocytes, with mild significance (*p* = 0.07, Supplementary Data 14), and positive correlation between neutrophil and monocyte proportions were only detected among controls upon EV infection (correlation coefficient 0.66 in controls (Fig. 3d, Supplementary Data 13a), 0.05 in cases (Fig. 3e, Supplementary Data 13b)). A similar albeit weaker positive correlation was observed both in cases and controls upon HAdV (0.19 in controls, 0.18 in cases) (Supplementary Fig. 16a, b), thus showing a clear difference between cases and controls only in EV infection. Reduced correlation between RNA and protein levels of immune response genes C2 and SERPING1 was also observed in EV+ cases (Spearman ρ = 0.05 and ρ = 0.07 for C2 and SERPING1, respectively), in relation to controls (Spearman ρ = 0.50 and ρ = 0.59 for C2 and SERPING1, respectively), further supporting the robust properly regulated host response to EVs among control children (Supplementary Fig. 13).

To gain insight into the variability of EV-induced responses in case children we partitioned the case samples further into distinct clusters based on expression profiles and performed correlation analysis (see Methods) with inflammatory episodes (fever, respiratory and gastrointestinal) and virome infections. We found that a subset of islet autoimmunity cases (7/38 samples, partitioned as the right cluster shown in Fig. 3a), representing a negative immune response, were positive only for CVB group EVs (types CVB2, CVB3, and CVB5), while the other islet autoimmunity cases were infected by a broader selection of EV types (Fig. 3f).

Overall, the results suggest that there is a clear and consistent immune response upon EV infections in autoantibody negative control children. This response is statistically distinct from the more diverse and aberrant response detected in children developing autoimmunity with observed differences with respect to monocytes, neutrophils, and CD8+ T cells.

## Discussion

In this study, we detected distinct temporal gene expression patterns and immune cell type proportions between islet autoimmunity cases and matched control children prior to the onset of the islet autoimmunity. Moreover, these differences were distinct in children with different types of IAbs. Our gene expression and cell type deconvolution data originates from whole blood samples. Even though blood samples can not fully recapitulate the disease processes taking place in

the pancreas they can serve as a non-invasive source for marker analysis and allows us to obtain information about the immune processes, potentially prior to the detection of islet autoimmunity in pancreatic samples. For example, the decrease of blood B cell proportion can also reflect increased cell recruitment to the pancreas or other inflamed tissues. The systemic nature of immune responses and the demonstrated ability to detect them from the blood supports analysis in this setting.

The discovery of T1D subtypes supports the heterogeneity of disease pathogenesis and offers completely new opportunities to dissect the mechanisms leading to T1D. From the histological point of view, distinct subtypes can be identified based on the pattern of immune cells that infiltrate the pancreatic islets, correlating with the age at diagnosis of T1D[50,51]. In this study we focused on the subtypes that were previously identified by characterizing the autoantigen specificity of the first-appearing IAb (either IAA or GADA)[7]. Based on our results, abnormalities in the function of the immune system emerge already long (9–12 months) before IAbs appear. Additionally, the immune cell type proportions clearly differed in the two main subtypes of islet autoimmunity, characterized by the IAA-first and GADA-first autoantibody patterns: there were more DE genes in the GADA-first than other groups and they were predominantly detected at 12 months, whereas most DE genes in IAA-first group were detected at 9 month. A similar pattern was detected for the genes in the conditional logistic regression analysis: we detected more significant transcriptomic changes in GADA-first children (14 genes) compared to IAA-first children (1 gene), and the associations mostly came from the window of 12 months prior to GADA seroconversion (also recently reported in ref. 24). In addition to gene expression changes, the patterns in cell type proportions were different for IAA-first and GADA-first. These findings emphasize the importance of these two subtypes as markers of two different pathogenetic pathways that can lead to islet autoimmunity and T1D.

The expression of *FABP5*, the fatty acid binding protein, was associated with increased risk of IAA-first (gene+HLA+SNPs, months 9 [OR 7.17 (1.40–36.52) *p*-value 0.02] and 12 [OR 2.70 (1.06–6.87) *p*-value 0.04] prior to seroconversion) but not GADA-first, being in line with previous report of the *FABP5* expression[52]. *FABP5* also has anti-inflammatory properties[36,38] and it contributes to the function of CD8+ tissue resident memory T cells regulating the pathogenesis of viral infections[53,54]. Our results suggest that the IAA associated autoimmune process may be different from GADA associated process, as cited previously in TEDDY and other studies[11,16,18,55].

We also reported a lowered islet autoimmunity risk in the full islet autoimmunity NCC1 (OR 0.59 (0.42–0.84) and GADA-first (OR 0.45 (0.25–0.82)) with higher eosinophil proportions at 12 months prior to islet autoimmunity. Biological significance of this finding is not known. Theoretically, it could reflect an increase in early age infections, particularly in parasitic infections, or it may be a reflection of an altered function of the immune system as seen e.g. in certain allergic diseases. Further studies are needed to find out whether this phenomenon reflects the activation of immune pathways that are induced by common infections and which may protect against islet autoimmunity[56]. Additionally, we found that the relative proportion of monocytes was increased especially in the IAA-first cohort (at 12 and 9 months), and decreased B cell proportions were associated with different timepoints prior to GADA (3 months) or IAA-first (6 and 9 months) islet autoimmunity. All these results indicated differences in the development of GADA or IAA-related autoimmunity while uncovering markers and targets for further mechanistic studies dissecting the etiopathogenesis of these two main T1D subtypes.

Another striking finding was related to the *GSTM1* expression. *GSTM1* heterozygosity has been linked to T1D[32]. We observed a bimodal distribution in *GSTM1* expression and higher expression

associated with a higher risk to GADA-first islet autoimmunity. We confirmed that this observed expression is attributed to the genetic alteration at the loci. This is consistent with a previous report showing that wild-type *GSTM1* is a risk factor for T1D, especially for cases with disease onset at the age of 14–20 years[32]. In addition, *GSTM1* polymorphism regulates the susceptibility to many viral infections, including e.g. chronic hepatitis B, severe COVID-19, tick-borne encephalitis and human papillomavirus infections[57–60].

Very few studies have reported transcriptomic differences within the context of islet autoimmunity and particularly multiple timepoints prior to the detection of IAbs. Recently and also within the TEDDY community, microarray based transcriptional signatures of enrichment of NK cells[24] and heightened inflammation along with irregular lipid metabolism[11] were found to be associated with islet autoimmunity while enrichment of B cells was seen in control children[24]. This aligns with our and previous findings of more robust immune responses[61] in the control children upon the initial EV infection, as well as with significant associations in decreased B cell proportions across IAb patterns (IAA-first and GADA-first) and timepoints. The integration of genetics with time-dependent transcriptomics, virome and immune cell type alterations prior to IAb seroconversion consistently outperformed sparser models suggesting that these integrated models may help to identify pathogenetic pathways. In addition, these models pave the way towards development of more accurate assays for early detection and eventual prevention of pathogenic processes leading to T1D.

Our findings confirm the previously documented association between EV-B infections and initiation of islet autoimmunity in TEDDY children[18]. In the present study we used a bioinformatic pipeline, based on Vipie[62], which can identify various viral taxa by assembling complex metagenomic sequence data and comparing it to reference sequences available in public databases. Moreover and further described within methods, this pipeline is optimized to identify viral subtypes (e.g. EV genotypes) using only those sequence reads that map to the genome region that codes the viral capsid proteins that include the viral subtype determinants. Thus, the fact this method again found CVB group viruses associated with islet autoimmunity supports the robustness of this finding.

Furthermore, when we analyzed sample pairs collected before and after infections by different EV types, we detected clear immune responses to EV infections in the islet autoimmunity negative control children, represented by an enrichment in upregulated innate immunity genes, while the response in islet autoimmunity positive case children was more variable. Thus, a robust antiviral response against EV may be a protective factor for islet autoimmunity, being present especially among children who did not develop islet autoimmunity later on. The mechanism of this protection is not known but it could be related to better immune defense against diabetogenic viruses. This is consistent with the observed association between EV infections and later development of islet autoimmunity in the TEDDY study[18] and in other studies[19,63]. Interestingly, the recent observation from the TEDDY study[18] showed that islet autoimmunity-associated EV infections were atypically prolonged. Thus, it is possible that the weak immune responsiveness to the virus may contribute to this phenomenon leading to delayed eradication of the virus. We also found that increased monocyte proportions, which can be caused by chronic or subacute infections, were associated with islet autoimmunity both in the full islet autoimmunity and IAA-first cohorts. A less pronounced immune response to EVs in islet autoimmunity cases was supported by HAdV analysis. For both viruses, less differentially expressed genes were detected in case children compared to control children, however, the difference was considerably more pronounced with EV infection.

In conclusion, our study showed immune related transcriptomic differences between case and control children prior to islet autoimmunity. This phenomenon is presented differently in children with either IAA or GADA as the first appearing IAbs. We also found that EV infections induce less robust antiviral response in children who later develop islet autoimmunity as compared to control children.

## Methods

### Sample processing and sequencing

Whole blood samples collected from children included in the TEDDY NCC1 islet autoimmunity cohort[9,17] were used for whole transcription sequencing. Blood sample collection begins at the 3 month study visit and continues every 3 months until the child is 4 years old (if persistent islet autoimmunity is developed, until 15 years old) otherwise switches to 6 month intervals until 15 years old[64]. The TEDDY study collected ~2.5 ml of whole blood for total RNA extraction using Applied Biosystems Tempus blood RNA tubes. The KingFisher 96 robotic system from Thermo Fisher and the MagMax magnetic bead technology in 96-well format were used for the high throughput extraction of RNA samples. Total RNA was extracted using the MegMax magnetic bead technology from frozen whole blood samples by the TEDDY RNA Laboratory at Jinfiniti Biosciences. The RNA samples were prepared using Illumina's TruSeq Stranded mRNA Sample Prep Kit. RNA-sequencing was performed, via 61 batches, on the Illumina HiSeq4000 platform with paired-end $2 \times 101$ bp reads with a targeted 50 million reads per sample by the Broad Institute, Cambridge, MA.

For virome analysis stool samples were collected monthly from 3 to 48 months of life, then every 3 months until the age of 10 years while plasma samples were collected every three months from 3 to 48 months of life, then every 6 months. Appropriate packages, shipping boxes and delivery schedules were provided. Details of the experimental design, sample collection and processing, and data generation of the TEDDY virome study have been published previously[18,65].

The TEDDY study was approved by local US Institutional Review Boards and European Ethics Committee Boards in Colorado's Colorado Multiple Institutional Review Board, Georgia's Medical College of Georgia Human Assurance Committee (2004–2010), Georgia Health Sciences University Human Assurance Committee (2011–2012), Georgia Regents University Institutional Review Board (2013–2015), Augusta University Institutional Review Board (2015–present), Florida's University of Florida Health Center Institutional Review Board, Washington state's Washington State Institutional Review Board (2004–2012) and Western Institutional Review Board (2013–present), Finland's Ethics Committee of the Hospital District of Southwest Finland, Germany's Bayerischen Landesärztekammer (Bavarian Medical Association) Ethics Committee, Sweden's Regional Ethics Board in Lund, Section 2 (2004–2012) and Lund University Committee for Continuing Ethical Review (2013–present). All parents or guardians provided written informed consent before participation in genetic screening and enrollment. The study was performed in compliance with all relevant ethical regulations. Study has been registered at clinicaltrials.gov with trial registration number NCT00279318.

### RNA-seq alignment and quality control

Raw RNA-sequencing data from the TEDDY samples was aligned to hg19 using STAR v2.4.1[66] software with the default parameters except for the following: --sjdbScore 2 --outSAMattributes NH HI NM MD AS XS --outFilterType BySJout --outSAMunmapped Within --outFilterScoreMinOverLread 0 --outFilterMatchNminOverLread 0 --outFilterMismatchNmax 999 --outFilterMultimapNmax 20. The resulting read counts were extracted with FeatureCounts v2.0.0[67]. Poor quality samples with unassigned reads proportion > 40% were filtered out from downstream analysis based on PCA visualizations. RNA-Seq quality scores (RQS) were assessed, with >90% exceeding RQS of 5.5 (median 7.54, IQR 6.67–8.16). In addition we found

similar normal RQS distributions between islet autoimmunity cases and controls, as shown in Supplementary Fig. 15. The cell type references and validation data was aligned to hg19 using STAR v2.5.3[66] with the same parameters as when processing the TEDDY samples.

## Harmonizing subject omic samples prior to islet autoimmunity seroconversion

Data harmony of virome and transcriptome samples are performed by matching on common months prior to seroconversion. TEDDY study uses a subject serial variable 'due num' which dictates the expected and sequential sample from enrollment. This 'due num' is used to match case and control subjects and the pairs with RNASeq as well as virome are harmonized and collected (relative from 12 months prior to seroconversion, counts listed in Supplementary Data 1a, b) for the conditional logistic regression analysis described above. After data harmonizing of matched case and control samples on TEDDY due month, virome profiling consists of 9072 stool samples (mean 9 per subject) and 4686 plasma samples (mean 5 per subject) analyzed for virome.

## Statistical testing to detect temporal differential expression patterns

Differential expression analyses of gene read count data were performed using DESeq2[68]. In each time point, DESeq2 (Wald test) was applied on raw count data (only protein coding genes and after filtering for low count genes) with a multi-factor design formula which includes the condition (islet autoimmunity case or control), sample pair and HLA information. Multiple-hypothesis testing was considered by using Benjamini–Hochberg correction[69]. For each time point, we selected genes with adjusted $p$-value < 0.05 and |LFC|>0.5. We combined all selected genes from different time points to form a set of differentially expressed (first step gene selection) genes.

## Temporal filtering of differentially expressed genes

Gene count data were normalized using the DESeq2[68] package, and then the resulting gene expression data were min-max scaled to have a minimum value of zero and maximum value of 1. We then calculated the median of each selected gene separately for case and control samples. A slope was defined as the difference of the median gene value between consecutive time points, resulting in a set of 4 slope values from time point seroconversion to 12 months prior to seroconversion, for each selected gene for case and control groups. Maximum slope values for each gene in cases were scaled between zero and one. Finally, genes with higher slope in cases were selected with criteria: normalized max slope in cases >0.15 and median slope in controls is less than half of the median slope in cases. To further adjust for testing across 4 timepoints, we reduce the reporting $p$-value threshold (derived by conditional logistic regression described below) from 0.05 to 0.01 or minimum of 2 timepoints with $p$-values ≤ 0.05. These criteria were set to highlight the genes, where the temporal effect is due to altered expression in cases, and thus, more likely disease relevant.

## Conditional logistic regression for virome infections, gene expression, and cell type proportions

Conditional logistic regression (CLR) was to assess the associated odds of virome infections and the expression of the selected genes and different cell types to the islet autoimmunity outcome in the NCC1 setting of the matched pair indicator identifier. The analysis with CLR is performed in R using the survival package (https://CRAN.R-project.org/package=survival) while adjusting for HLA-DR/DQ genotype. Multiple comparison adjustments are not applied due to nested-case control settings.

## Model performance

Model performance was compared using $p$-values from Likelihood ratio tests on all selected genes and months 3, 6, 9, and 12 prior to islet autoimmunity onset. We compute the baseline with gene only, hla and SNPs (*INS* and *PTPN22*), and iteratively adding gene and virus variables to the models.

## Deconvolution analysis and benchmarking

Validation data set that contained five RNA sequenced whole blood samples with known cell type fractions was obtained from Gene Expression Omnibus (GEO accession GSE60424). Expression profiles for cell types present in whole blood were obtained from NCBI Sequencing Read Archive (Supplementary Data 15).

For deconvolution analysis, all datasets (TEDDY samples, validation samples, cell type samples) were first normalized with DESeq2, then quantile normalized together with R package preprocessCore and lastly transformed to logarithmic scale. Replicates of cell types were combined with median. A reference sample representing a typical whole blood sample was formed by taking a median across 190 control subject samples selected at timepoints that matched to their NCC1 case seroconversion months that were not included in the further analyses (Supplementary Data 1d).

To estimate the relative proportions of cell types, we used regression analysis with elastic net regularization as described in detail in refs. 70,71. In short, regression analysis was used to model the whole blood expression profile as a combination of the expression levels of individual genes from different cell types and the reference sample, weighted by the respective cell type proportions. Weights for cell type proportions were estimated using elastic net regularization. These weights reflect the contribution of each cell type on top of the reference sample profile in explaining the observed whole blood expression profile. Regression was performed using all the expressed genes ($n = 17,964$) with elastic net mixing parameter $\alpha = 0.25$.

For validation of the model, the reference sample was made with all the five healthy control subjects' samples (GEO accession GSE60424) and regression was performed separately using the same parameters. Pearson correlation with the ground truth cell type proportions was 0.920 (Supplementary Fig. 9).

## Virome profiling and capsid genotyping

We performed virome profiling using a custom version of Vipie[62] and further genotyping of viruses by mapping the sequences on EV capsid proteins (VP1, VP2, VP3 and VP4) and HAdV capsid proteins (hexon, penton and fiber) of virus strains available from Tampere Virology group and GenBank[72]. Samples detected by Vipie, with a minimum of 5 hits of sequence reads with viral reference sequences, were selected for capsid mapping via BWA[73] using optimally selected contigs assembled by Velvet[74] and SPAdes[75] and further matched on minimum MAPQ of 30 (.999 probability). Discordance between case and control number of virus exposed samples was tested using conditional logistic regression. We defined consecutive EV infections as two or more samples positive for different EV genotypes in the same child, while viral persistence as reported[18], could not be obtained due to the shorter capsid regions applied by Vipie on genotyping.

## Virome impact on host transcriptomic profiles

Using the EV and HAdV genotype mapped results, we matched the following first RNASeq sample available, up to 3 months after the first EV or HAdV infection as stool virome samples are collected monthly while whole blood sample is taken quarterly. The matching control sample, selected from the same host, is the prior infection free sample taken prior to the virus infection. The sample selection steps are repeated for EV and HAdV initial infections on control children. In this setting, our data set yielded 44 islet autoimmunity case sample pairs

(including 5 EV detections from plasma virome) and 13 control sample pairs (including 2 EV detections from plasma) with available transcriptome data. For HAdV we identified 88 case sample pairs (26 from plasma) and 83 control sample pairs (20 from plasma).

## Correlation between interested viral responses and inflammation episodes

Using results from EV impact on host transcriptomic profiles, we labeled case subjects based on hierarchical clusters partitioned from differentiated immune genetic genes (high, medium and low) and viral (EV and HAdV) groups and their strains of interests (HAdV-F, CVB1-5). Correlation maps are plotted using corrplot[76] and statistical testing (*p*-value < 0.05) performed using Pearson testing in R.

## GSTM1 genotyping

Whole genome sequencing (WGS) data, produced using the Illumina HiSeq X Series platform with paired-end 2 × 150 bp reads by Macrogen (USA), from a subset of the NCC1 RNA-seq cohort (345 pairs, 83%) was used to determine the genotypes of *GSTM1* gene. The raw sequences were aligned to the Genome Reference Consortium Build 38 (GRCh38DH) based on Burrows–Wheeler Aligner (BWA) as previously described[77]. WGS data was analyzed by TEDDY data coordinating center using an in-house version of the Trans-Omics for Precision Medicine (TopMed) GotCloud pipeline[78,79]. As deletion of the gene has been reported previously, we obtained the read counts from the loci of the *GSTM1* gene using Mosdepth -tool version 0.3.3[80].

## Protein data analysis

Targeted mass spectrometry based proteomics data measured from the plasma samples of TEDDY NCC1 cohort was used. These data were generated and processed as described earlier (Nakayasu et al. 2022). Log2 fold changes (LFC) of 167 proteins between cases and controls from the TEDDY NCC1 cohort were downloaded (Nakayasu et al. 2022). These data results are from the same experimental design, and thus are comparable to our RNA-seq data at the LFC level. To access pathway enrichment concordance, Reactome database pathway enrichments were evaluated with Enricher tool[27]. Two genes, C2 and SERPING1, from our EV analysis matched the set of 167 proteins. Peptide level protein data from each sample was provided by the TEDDY data coordinating center. Median of all individual peptides from each protein was calculated and Spearman correlation was applied to evaluate the correlation between RNA and protein levels in EV detected cases and controls separately.

## Reporting summary

Further information on research design is available in the Nature Portfolio Reporting Summary linked to this article.

## Data availability

The TEDDY sequencing data used in this study have been deposited in the dbGaP database under accession code phs001442.v4.p3. The TEDDY omics data are available under restricted access for sensitivity reasons, access can be obtained by request through dbGaP. The results data including log fold changes and cell type coefficients generated in this study are provided in the Supplementary Information/Source Data file. The mass spectrometry raw data used in this study are available in the MassIVE database under accession codes MSV000091560 (untargeted proteomics) and MSV000091562 (targeted proteomics) [https://massive.ucsd.edu/]. Clinical metadata analyzed for the current study is available in the NIDDK Central Repository [https://repository.niddk.nih.gov/studies/teddy/]. The deconvolution validation data used in this study is available in the Gene Expression Omnibus (GEO) under accession code GSE60424. The reference cell type data used in deconvolution analysis are available in the GEO database under

accession codes GSM971331, GSM823383, GSM1060237, GSM3319903, GSM1576438, GSM986103, GSM996197, GSM996200, GSM3039712, GSM3039716, GSM3039720 and GSM1657640. Source data are provided with this paper.

## Code availability

Analysis was performed using open source tools and libraries. Custom scripts for key analysis steps are available on the GitHub repository https://github.com/NykterLab/TEDDY_IA (https://doi.org/10.5281/zenodo.8345041). Vipie virome web application scripts are available on https://sourceforge.net/projects/vipie/ and hosted on http://vipie.rd.tuni.fi/vipie/index.html. The enterovirus capsid libraries are available upon request from the Tampere Virology Group (https://research.tuni.fi/virology/).

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

## Acknowledgements

The TEDDY Study is funded by U01 DK63829, U01 DK63861, U01 DK63821, U01 DK63865, U01 DK63863, U01 DK63836, U01 DK63790, UC4 DK63829, UC4 DK63861, UC4 DK63821, UC4 DK63865, UC4 DK63863, UC4 DK63836, UC4 DK95300, UC4 DK100238, UC4 DK106955, UC4 DK112243, UC4 DK117483, U01 DK124166, U01 DK128847, and Contract No. HHSN267200700014C from the National Institute of Diabetes and Digestive and Kidney Diseases (NIDDK), National Institute of Allergy and Infectious Diseases (NIAID), Eunice Kennedy Shriver National Institute of Child Health and Human Development (NICHD), National Institute of Environmental Health Sciences (NIEHS), Centers for Disease Control and Prevention (CDC), and JDRF. This work is supported in part by the NIH/NCATS Clinical and Translational Science Awards to the University of Florida (UL1 TR000064) and the University of Colorado (UL1 TR002535). The content is solely the responsibility of the authors and does not necessarily represent the official views of the National Institutes of Health. We acknowledge additional financial support from the Academy of Finland (project no. 312043, M.N., 325999, K.J.R.), European Commission Horizon 2020 programme (grant no. 825033, M.N.), The Diabetes Research Foundation (M.N.), The Leona M and Harry B Helmsley Charitable Trust (H.H.), Sigrid Juselius Foundation (M.N., K.J.R., H.H.), and Päivikki and Sakari Sohlberg's Foundation (J.E.L.).

## Author contributions

J.L., J.P.K., J.T., Å.L., J.F.P., N.J.A., J.X.S., W.A.H., M.J.R., R.E.L., K.J.R., H.H., and M.N. designed the study. J.T., Å.L., M.J.R., W.A.H., J.X.S., H.H., and J.P.K. participated in patient recruitment and diagnosis, sample collection, and generation of the metadata. K.S., E.M., T.H., S.R., and H.M.P. processed RNA sequencing data. J.L., T.H., J.F.P., N.J.A. and R.E.L. generated and processed the virome sequencing data. J.L., E.M., K.S., S.L., S.R. and H.M.P. performed the data analysis and figure generation. J.L., E.M., K.S., J.E.L., S.O., K.J.R., H.H. and M.N. performed the data interpretation. J.L., E.M., K.S., K.J.R., H.H. and M.N. wrote the paper. All authors contributed to critical revisions and approved the final manuscript.

## Competing interests

H.H. is a shareholder and chairman of the board of Vactech, and a member of the Scientific Advisory Board of Provention Bio, which develops vaccines against picornaviruses and CVB. The remaining authors declare no competing interests.

## Additional information

[1]Prostate Cancer Research Center, Faculty of Medicine and Health Technology, Tampere University and Tays Cancer Centre, Tampere, Finland. [2]Biostatistics, Health Sciences, Faculty of Social Sciences, Tampere University, Tampere, Finland. [3]Finnish Institute of Molecular Medicine, FIMM, University of Helsinki, 00290 Helsinki, Finland. [4]Department of Medical Epidemiology and Biostatistics, Karolinska Institute, Stockholm, Sweden. [5]A.I. Virtanen Institute for Molecular Sciences, University of Eastern Finland, Kuopio 70150, Finland. [6]Department of Virology, Faculty of Medicine and Health Technology, Tampere University, Tampere, Finland. [7]Health Informatics Institute, Morsani College of Medicine, University of South Florida, Tampa, FL, USA. [8]Research Centre for Integrative Physiology and Pharmacology, Institute of Biomedicine, and Centre for Population Health Research, University of Turku, Turku, Finland. [9]Department of Pediatrics, Turku University Hospital, Turku, Finland. [10]Department of Clinical Sciences, Lund University CRC, Skåne University Hospital, Malmö, Sweden. [11]Alkek Center for Metagenomics and Microbiome Research, Department of Molecular Virology and Microbiology, Baylor College of Medicine, Houston, TX, USA. [12]Platform for Innovative Microbiome & Translational Research (PRIME-TR), Moon Shots™ Program, The University of Texas MD Anderson Cancer Center, Houston, TX, USA. [13]Jinfiniti Precision Medicine, Inc., Augusta, GA, USA. [14]Pacific Northwest Research Institute, Seattle, WA, USA. [15]Department of Medicine, University of Washington, Seattle, WA, USA. [16]Barbara Davis Center for Childhood Diabetes, University of Colorado, Aurora, CO, USA. [17]Fimlab Laboratories, Tampere, Finland. [18]Foundation for the Finnish Cancer Institute, Helsinki, Finland. [19]These authors contributed equally: Jake Lin, Elaheh Moradi, Karoliina Salenius. ✉e-mail: kirsi.rautajoki@tuni.fi; heikki.hyoty@tuni.fi; matti.nykter@tuni.fi

## the TEDDY Study Group

**Jutta E. Laiho**[6]**, Sami Oikarinen**[6]**, Hemang M. Parikh** ⬡ [7]**, Jeffrey P. Krischer**[7]**, Jorma Toppari** ⬡ [8,9]**, Åke Lernmark** ⬡ [10]**, Joseph F. Petrosino**[11]**, Nadim J. Ajami** ⬡ [11,12]**, William A. Hagopian** ⬡ [14,15]**, Marian J. Rewers** ⬡ [16]**, Richard E. Lloyd**[11] **& Heikki Hyöty** ⬡ [6,17] ✉

