## [Peer Review File · Nature Communications]

Distinct transcriptomic profiles in children prior to the appearance of type 1 diabetes-linked islet autoantibodies and following enterovirus infectionREVIEWER COMMENTS

Reviewer #1 (expert in type-1 diabetes and β -cell transcriptomics):

In the present study the Authors utilized longitudinal whole blood transcriptomic sequencing data to search for putative host responses during the follow up of children ahead of the first appearance of islet autoantibodies. They next correlated the observed response profiles with disease autoantibody patterns and indications of enterovirus infection, suggesting a clearer virus-induced immune response in autoantibody negative children as compared to more variable responses in children that eventually evolved to develop islet autoimmunity. These findings are interesting but are limited by the nature of the transcriptomics (see below), and it is difficult to discern clear mechanistic information from the data.

1. RNAseq was performed in whole blood and it is uncertain whether this provides sufficiently precise information on the relevant pathogenic process taking place the islet level. Indeed, as immune cells leave the circulation to home in the islets this may be detected in whole blood as a decrease instead of an increase in some cell types or mRNAs.
2. The supplementary tables are not sufficiently self-explanatory. Abbreviations should be defined and an explanatory paragraph provided at the bottom of the tables.
3. Besides some expected correlations, such as an increase in innate immunity markers in the blood of viral-infected individuals (which seems somewhat less marked in individuals who developed islet autoimmunity), it is difficult to infer mechanistic conclusions from the data shown.
4. It is mentioned in the Discussion that "...Another striking finding was related to GSTM1 expression...We observed a bimodal distribution in GSTM1 expression, that can putatively be a proxy for genetic alterations at the loci". This hypothesis should have been validated by genotyping the individuals studied.

Reviewer #2 (expert in type-1 diabetes):

This is an analysis from the TEDDY study that presents data from whole blood transcriptomic analysis performed in the nested case-control cohort described in many papers from the study group. The case control cohort is comprised of children who do and do not develop islet autoimmunity during longitudinal follow-up beginning at birth. For this study, RNA sequencing was performed using whole blood collected longitudinally. Results from whole blood transcriptomic analysis between children with or without IA are presented and then changes are also described within the insulin aab first and GADA aab first groups. The authors also examine transcriptomic changes in association with viral infections, comparing to previously published analyses. An obvious strength of the study is that it includes analysis from a cohort that is incredibly unique and unparalleled in the degree of phenotyping and longitudinal follow-up. A weakness of this report is that the TEDDY dataset and in particular this paper are dense. As a reader, it was hard to settle on the main point. In addition, some of the data is confirmatory of other papers from the TEDDY study, and there is no real validation of results. Listed below are additional major and minor concerns:

1. What type of collection tube was used for RNA samples? Was the collection method consistent across all sites, and were all samples analyzed together as batch effects can sometimes complicate analysis in multi-center trial efforts
2. Information is not provided on the basic demographics of the study populations, either in the main text nor in the supplemental material. This is really needed to contextualize findings.
3. An interesting question is what differences exist between the IA and GADA first groups. However, these groups are not really compared. This might speak to true differences in subtypes, which is presented as a concept but never really fully developed by the authors
4. In Figure1a, microbiome data is referred to, but this data is not mentioned in the following

analysis and results.

5. Figure 2b, there are not dramatic differences observed in the Vlnplots. The authors should show fold change instead of normalized gene expression.

6. Heatmaps and Vlnplot are utilized in both the main supplementary figures. However, this is not a very straightforward way of presenting differences, especially in a longitudinal trajectory analysis. It is hard from these figures to appreciate a difference across time.

7. In the deconvolution analysis, is the reference from a control population (and by corollary what are clinical characteristics of that population). How was the regression analysis performed in the deconvolution. This method is not very clearly described and this analysis seems somewhat tangential to the overall message.

8. What is the reference of OR detected in Figure 2a, c. It should be labeled and mentioned.

9. I am puzzled by the prediction data and analysis. What exactly are the authors trying to predict, and how would such a model ever be used? For example, why would you need 4 pieces of high dimensional data to answer the question of whether there is an antibody (which can be easily measured). Some additional context or justification for this analysis should be provided. Also, it would seem any prediction work might come at the end of the paper.

10. The results and discussion are lengthy, and not very well organized. In part this seems to be the case because there does not seem to be an overarching message. This vagueness is reflected in the abstract, which requires more specificity when describing the goal, results, and interpretation.

11. The authors don't really introduce the importance of HAdV infections before discussing integration into the analysis

12. The label of Figure 3 is not very clear and is confusing. For figure 3 d,e, what does each column and row represent? These should be labeled.

13. The authors suggest that altered GSTM1 expression could be a proxy for genetic risk. This seems quite speculative. Moreover, don't they have genetic data such that this notion could be tested?

Reviewer #3 (expert in viral aetiology of endocrine disorders):

This is an important, complex and somewhat difficult-to-read contribution. Conclusions are of interest. Children prone to type-1 diabetes – when followed longitudinally for long times – show transcription profiles in peripheral blood that differ from those of non-diabetes-prone children not developing pancreatic autoantibodies. The altered transcription profiles refer especially to innate immunity and reveal that diabetes-prone children have a somewhat attenuated immune response to enterovirus infection (but not so much to adenovirus infection).

The study attempts to integrate genetics, transcriptomics over prolonged times, and immune cell type alterations over time with the detection of two virus groups (enterovirus and adenovirus) at serial time points before the appearance (seroconversion) of autoantibodies to pancreatic islet antigens (insulin, GADA).

The MS contains text with references and figures plus Suppl Materials with Tables and Figures, plus Reporting Summary.

Title: the title is not clear to me.

I may suggest: "TEDDY Study of type 1 diabetes: peripheral blood transcriptomics reveal two distinct pathogenic paths associated with the autoimmune response, virus detection, and the antiviral response"

Abstract

In the present form it does not entirely reflect the results and, especially, the main conclusions as expressed in the Discussion. It should be rewritten.

Samples for transcriptomics (probably frozen whole blood) and for virus detection need to be indicated. Methods say that stool is one sample (in Methods it is not mentioned how stool was collected and stored). The second sample is said to be serum (abstract and elsewhere in text). Probably, the sample is plasma OR whole blood as for transcriptomics. Please clarify. The type of sample needs to be specified all over the text (also in methods) and in Figures.

Findings are innovative and derived from extensive studies. The main conclusion is that children

genetically predisposed to type 1 diabetes seem to have an attenuated and partially ineffective antiviral response (mainly to enteroviruses) that could favor virus persistence, chronic inflammation, and the production of autoantibodies to insulin and/or GADA. The findings succeed in enlightening the recognized relation between genetic and environmental factors in the origin of type 1 diabetes. In addition, they confirm the already known pathogenic heterogeneity within type 1 diabetes.

Thus this contribution may have notable implications in translational medicine.

Some points need clarification:

Based on conversion to autoantibody production, two subgroups of type 1 diabetes have been studied: a) first insulin, b) first GADA. The groups should be indicated IA-first and GADA-first (all over the text, Tables and Figures)

Line 69: stool – as previously indicated, please insert collection and storage. Then: why stool? In fact, previous work from the Authors' group showed that virus in stool does not predict autoantibody seroconversion, nor development of clinical diabetes. Probably, the sample was available and has been studied. However, no mention of stool results appears in Results or Discussion. If no important data have been obtained from studies, I'm suggesting to delete this specimen.

Line 88: these environmental... It is not clear to me which determinants. Please rephrase.

Line 103: data of cohorts. The study design is difficult to follow. I may suggest to insert Table 1 (Table S1 currently in Suppl Materials) with cases and controls. NCC: please define.

Figure 1: rather complex to follow the numbers: 418 (1:1), then 383 Microbiome and 370 IA (full?) transcriptome. The stool, and (again) plasma (? , whole blood?), then RNAseq (is it whole blood?)

Line 145: using HLA status covariate. Unclear. Rephrase for clarity.

Figure 2: Full IA TAS2R30 not considered. GADA first, only GSTM1 considered. Why? Please comment at least in the legend.

Discussion is well organized. Since conclusions are numerous, I would suggest to number them 1 to 8: abnormalities of the immune system; two different pathways; results consistent with the association of prolonged or cumulative EV infection with IA production; defects of antiviral responses more relevant to EV compared to HAdV; confirmed relevance of GSTM1 to type 1 diabetes; CVB apparently more relevant to autoimmunity than other EV genotypes; possible role of eosinophil profile; increase of monocytes and decreased B cell proportions.

Line 425: Clarify whole blood (Na-EDTA or K-EDTA)? Then, detection of adenoviruses: was the assay based on RNA (virus transcripts), or DNA (virus genome)? If DNA, please add how it was extracted and prepared for NGS:

Line 450: "serotyping": it is not serotyping, it is genotyping. Change all over the text.

References: adequate.

Legend to Figures (main and supplementary): adequate

Reporting Summary: OK

Response letter

We thank the reviewers for supporting comments about our work. We provide detailed responses to the raised criticism below and have revised the manuscript accordingly.

REVIEWER COMMENTS

Reviewer #1 (expert in type-1 diabetes and β -cell transcriptomics):

In the present study the Authors utilized longitudinal whole blood transcriptomic sequencing data to search for putative host responses during the follow up of children ahead of the first appearance of islet autoantibodies. They next correlated the observed response profiles with disease autoantibody patterns and indications of enterovirus infection, suggesting a clearer virus-induced immune response in autoantibody negative children as compared to more variable responses in children that eventually evolved to develop islet autoimmunity. These findings are interesting but are limited by the nature of the transcriptomics (see below), and it is difficult to discern clear mechanistic information from the data.

Response: We thank the reviewer for a supportive summary of our work. We have included additional analysis in our revision to address the raised concerns. We validate genetic regulation of GSTM1 expression with whole genome sequencing data. Furthermore, we have included protein level data to support our findings. In a parallel to our study, still unpublished study (Nakayasu et al. preprint available at: <https://doi.org/10.1101/2022.12.07.22283187>), TEDDY project has generated targeted plasma proteomics data from the same nested case control samples whose whole blood has been used for the transcriptomics analysis. As TEDDY has clear principles of data use in different manuscripts, comprehensive analysis of these data is not possible and would be outside of the scope of the current manuscript. Still, we are able to utilize these data for validation of our specific results. It should be noted that a number of published proteo-genomic studies have shown that overall correlation between transcriptome and proteome is not strong (Sinitcyn et al. Nat Biotech. 2023 23 Mar, Yang et al. Cell Syst. 2020 Aug 26;11(2):186-195.e9, Latonen et al. Nat Commun. 2018 Mar 21;9(1):1176.). Thus, expected correlation between any individual gene between RNA and protein level is low. More importantly, in our case, plasma proteome and whole blood transcriptome represent a different collection of molecular species. RNA level data represents mostly cellular RNA from leukocytes while plasma proteome represents secreted proteins and also other cell types, such liver cells, contribute to their secretion. TEDDY targeted plasma proteome data include 167 proteins, selected for analysis from a large scale protein analysis of a subset of patients. When compared to differentially expressed genes identified in our analysis only 7 genes are common between datasets: APOA2, C1QC, C2, IGFBP2, KNG1, SELENOP and SERPING1. As expected by the literature, we observe variable correlations between RNA and protein level, ranging from -0.159 for SELENOP to 0.161 for IGFBP2 when evaluated across all the samples in the cohort. Based on earlier studies, we hypothesized that even if the datasets do not match well at the level of

individual genes or proteins, we could observe convergence at the level of the pathway activation. Indeed, when performing pathway analysis using parallel lists of differentially expressed genes and proteins, we observe complement activation related pathways among the ones with strongest enrichment score in both datasets; while RNA level analysis also reveals a number of intracellular pathways only observed at RNA level, as expected (Supplementary Table 3). Two of the seven genes listed above, namely C2 and SERPING1, were found in the gene signature that was upregulated after enterovirus infection in control children (Figure 3a). These are both innate immunity related genes that are secreted and play a role in the complement system in plasma. We confirm that the expression change in protein level is consistent with RNA level change when compared in the same setting (Supplementary Figure 13). Furthermore, when analyzing correlation between RNA and protein level in case and control children with enterovirus infections, we observe strong correlation ($\rho > 0.50$) in controls that is lost in case children ($\rho < 0.1$) for both genes. This supports our key observation from transcriptome analysis, that control children who do not develop islet autoimmunity are exhibiting a robust (properly regulated) host immune response to enterovirus exposure. We hope that incorporation of these additional data will address the concerns regarding the reliance on transcriptomic data.

1. RNAseq was performed in whole blood and it is uncertain whether this provides sufficiently precise information on the relevant pathogenic process taking place the islet level. Indeed, as immune cells leave the circulation to home in the islets this may be detected in whole blood as a decrease instead of an increase in some cell types or mRNAs.

Response: We agree with the reviewer on this view on how the immune responses can be observed from the blood, fully acknowledging the indirect nature of the readout in terms of the effects taking place in the tissue. We have also added a notion about this into the discussion (page 10 second paragraph). However, blood also serves as a valuable and informative sample type which is less invasive than tissue samples to withdraw, and it is frequently used for biomarker detection. Collection of pancreas tissue samples is particularly challenging due to the high risk of complications. Therefore it is practically impossible to carry out among healthy individuals including children prospectively followed from birth. To increase robustness of our observations, we now also report protein level evidence, derived from plasma samples from the same cohort (see above).

2. The supplementary tables are not sufficiently self-explanatory. Abbreviations should be defined and an explanatory paragraph provided at the bottom of the tables.

Response: We have improved the tables and added definitions of the abbreviations and more detailed explanatory text to the tables.

3. Besides some expected correlations, such as an increase in innate immunity markers in the blood of viral-infected individuals (which seems somewhat less marked in individuals who developed islet autoimmunity), it is difficult to infer mechanistic conclusions from the data shown.

Response: We do acknowledge the descriptive nature of the work. However, we think that our results increase the understanding of islet autoimmunity development in the context of type 1 diabetes. For example:

- Different sets of differentially expressed genes and altered cell type proportions with distinct temporal patterns were observed in children with the IAA-first or GADA-first appearing autoantibody, which suggests that the path to IAA and GADA response differs at the molecular and cellular level.
- We observed increased monocyte and decreased B cell proportions as early markers occurring 9-12 months prior to autoantibody positivity.
- Genetically driven expression of *GSTM1* was associated with GADA-first autoantibody positivity.
- The integration with enterovirus infections demonstrated stronger individual variation in virus-induced immune responses in children who later developed islet autoimmunity whereas a more consistent antiviral response was detected in control children.

We have revised the text and abstract to highlight these key findings more clearly. In addition we have included genome and protein level data to support these findings (see the Manuscript main Results, page 5, first paragraph (Pathway Enrichment validation between transcriptomics and proteomics, Supplementary Table 3), page 8 second paragraph (Proteomics validation of host responses upon Enterovirus exposure) and supporting comparison plots of RNASeq and Proteomics host response marker LFC values in case and control subjects shown in Supplementary 13).

4. It is mentioned in the Discussion that "...Another striking finding was related to GSTM1 expression...We observed a bimodal distribution in GSTM1 expression, that can putatively be a proxy for genetic alterations at the loci". This hypothesis should have been validated by genotyping the individuals studied.

Response: We thank the reviewer for pointing this out. We have included genotype information from whole genome sequencing data (See Results, page 5, third paragraph and Methods, page 17, *GSTM1 genotyping* section) and demonstrate that 1) *GSTM1* expression is regulated by gene dosage effect, and 2) risk association of *GSTM1* gene to GADA-first autoantibody positivity holds also at genotype level. Data is shown in new Figure 2b and Supplementary Figure 8. In addition, our *GSTM1* findings are highlighted within Discussions (page 11 second paragraph).

Reviewer #2 (expert in type-1 diabetes):

This is an analysis from the TEDDY study that presents data from whole blood transcriptomic analysis performed in the nested case-control cohort described in many papers from the study group. The case control cohort is comprised of children who do and do not develop islet autoimmunity during longitudinal follow-up beginning at birth. For this study, RNA sequencing

was performed using whole blood collected longitudinally. Results from whole blood transcriptomic analysis between children with or without IA are presented and then changes are also described within the insulin aab first and GADA aab first groups. The authors also examine transcriptomic changes in association with viral infections, comparing to previously published analyses. An obvious strength of the study is that it includes analysis from a cohort that is incredibly unique and unparalleled in the degree of phenotyping and longitudinal follow-up. A weakness of this report is that the TEDDY dataset and in particular this paper are dense. As a reader, it was hard to settle on the main point. In addition, some of the data is confirmatory of other papers from the TEDDY study, and there is no real validation of results. Listed below are additional major and minor concerns:

Response: We thank the reviewer for a nice summary of our study. We have revised the manuscript structure with the aim to make the presentation less dense to read and highlight the key findings more clearly already in the abstract. We have added genome and plasma proteome data to validate key findings (see above).

1. What type of collection tube was used for RNA samples? Was the collection method consistent across all sites, and were all samples analyzed together as batch effects can sometimes complicate analysis in multi-center trial efforts

Response: Applied Biosystems Tempus blood RNA tubes were used as the collection tube and same protocol was used across all sites, as documented in TEDDY Manual of Operations (https://teddy.epi.usf.edu/documents/TEDDY_MOO.pdf). RNA samples were sent for the sequencing provider in 61 batches, with matching case and control samples included in the same batch to mitigate batch effect. Same versions of sample processing and sequencing protocols, library kits and sequencing instruments were used for the full cohort. These details have been included in the Methods section. After data normalization, we observed no significant biases that would result from batch effect, even when considering expression levels directly (i.e. without comparison to matched control from the same batch).

Response Figure 1: Principal component analysis of TEDDY transcriptome sequencing dataset. Samples from different batches have been indicated with different colors.

2. Information is not provided on the basic demographics of the study populations, either in the main text nor in the supplemental material. This is really needed to contextualize findings.

Response: The following demographics table, detailing TEDDY Nested Case Control (NCC1) islet autoimmunity study has been added to the Supplementary Table S1a and cited on page 4, second paragraph of the main text:

Response Table 1: Characteristics of the TEDDY NCC1 Islet autoantibodies (IA) (418 pairs) children matched on family history of T1D, country site, gender and age¹. Islet autoimmunity conversion onsets are the appearances of one or more islet autoantibodies (IAbs) (to insulin (MIAA), GAD65 (GADA) or IA-2 (IA2A)) confirmed at two consecutive visits.

Characteristics	N (Female (% or min-max))
Children	836 (44%)
Country	
Finland	228 (44.7%)
Germany	74 (48.6%)
Sweden	286 (41.3%)
USA	248 (45.2%)

Nonwhite	303 (41.9%)
Islet Autoantibodies (IA)	
GADA onset	301 (46.1%)
GADA onset mean age in days	854 (90-2084)
IA2A onset	170 (38.8%)
IA2A onset mean age in days	999 (79-2478)
MIAA onset	317 (43.5%)
MIAA onset mean age in days	684 (92-2313)
T1D onset	95 (45.3%)
T1D onset mean age in days	1021 (316-2290)

1. Lee HS, Lynch KF, Krischer JP; TEDDY Study Group. Nested case-control data analysis using weighted conditional logistic regression in The Environmental Determinants of Diabetes in the Young (TEDDY) study: A novel approach. *Diabetes Metab Res Rev.* 2020;36(1):e3204. doi:10.1002/dmrr.3204

3. An interesting question is what differences exist between the IA and GADA first groups. However, these groups are not really compared. This might speak to true differences in subtypes, which is presented as a concept but never really fully developed by the authors

Response: We thank the reviewer for this comment. In all the analysis we present, the comparison is indeed done within the nested case control setting and thus each case is directly compared to matched control. Direct comparison of GADA vs IAA groups would be problematic as these groups are known to have confounding differences, for example, different age of IA. Thus, we believe that the most informative means of comparison within the TEDDY cohort design is to compare the differences found in separate analysis of GADA and IAA groups. However, we now more clearly discuss the differences between the results obtained from IAA-first and GADA-first analysis.

4. In Figure1a, microbiome data is referred to, but this data is not mentioned in the following analysis and results.

Response: We thank the reviewer for pointing this out. We have now changed the wording throughout. Instead of microbiome, now systematically use the term virome as that is the only dimension of the data that we analyze.

5. Figure2b, there are not dramatic differences observed in the Vlnplots. The authors should show fold change instead of normalized gene Expression.

Response: We thank the reviewer for the comment. Benefit of the Vlnplots is that we can illustrate the full complexity of the dataset for the reader. But we fully appreciate the comment that we should highlight the differences more clearly. Thus, we have replaced the Vlnplots from the main figures with line plots of mean expression (with error bars) for case and control groups to more clearly visualize the expression difference between the groups. All the Vlnplots are now presented in the supplement to allow the detection of data distribution.

6. Heatmaps and Vlnplot are utilized in both the main supplementary figures. However, this is not a very straightforward way of presenting differences, especially in a longitudinal trajectory analysis. It is hard from these figures to appreciate a difference across time.

Response: As discussed in above reply, we feel that there is a value in providing a comprehensive view to the data by Vlnplots. With heatmaps we also aim to provide a more comprehensive view on the data than what would be possible by plotting individual genes. With these plots, we indeed aim to emphasize more of the trends than specific differences in the data. As we do provide details of log fold changes and p-values in Supplementary Tables, we hope the reviewer finds this approach acceptable.

7. In the deconvolution analysis, is the reference from a control population (and by corollary what are clinical characteristics of that population). How was the regression analysis performed in the deconvolution. This method is not very clearly described and this analysis seems somewhat tangential to the overall message.

Response: Reference is formed by taking a median across 191 control subjects from TEDDY study samples selected at timepoints that matched to their NCC1 case seroconversion months that were not included in the further analyses. We have included the characteristics of this population in Supplementary Table S1b. Details of the deconvolution analysis have been published earlier (Luoto *et al.*, 2018). However, this analysis is now described in more detail in methods -section. We have improved the text and expanded, for example pathway analysis, to make the relevance of the cell type deconvolution analysis more clear in the context of other analysis presented.

Response Table 2: Characteristics of the control children matched to their respective cases at the timepoint of seroconversion, which were used as the deconvolution median reference sample.

Characteristics	N (Female % (min-max))
Children	191 (46%)
Country	
Finland	64 (38%)

Germany	16 (56%)
Sweden	54 (46%)
USA	57 (53%)
Nonwhite	55 (47%)
Mean age in days	813 (182-2007)

8. *What is the reference of OR detected in Figure2a, c. It should be labeled and mentioned.*

Response: Figure 2a,c shows the Odds Ratios (ORs) resulting from conditional logistic regression analysis performed on differentially expressed genes (a) and cell types (c) using HLA as a covariate. With $OR > 1$, the respective gene / cell type is considered as positively associated with islet autoimmunity (a risk factor) whereas $OR < 1$ has a negative association (is protective). $OR = 1$ refers to the gene / cell having no association to IA. We thank the reviewer for pointing out the lack of labeling and have now included OR to the figure legend to clarify this point.

9. *I am puzzled by the prediction data and analysis. What exactly are the authors trying to predict, and how would such a model ever be used? For example, why would you need 4 pieces of high dimensional data to answer the question of whether there is an antibody (which can be easily measured). Some additional context or justification for this analysis should be provided. Also, it would seem any prediction work might come at the end of the paper.*

Response: We have clarified this section in the manuscript based on the comment. Main purpose of the analysis is the test, if transcriptomic markers that we identify in the analysis contain information that is independent of the established markers that are shown to increase risk of diabetes. By using logistic regression model to predict autoantibody positivity, we can quantify if incorporation of transcriptomic data improves the prediction accuracy. In addition, as transcriptomic data is dynamic, we can quantify the increase in accuracy at each time point. We are not proposing a model that would be readily applicable for clinical use, but our analysis does suggest that incorporation on specific transcriptomic markers (e.g. by PCR assay) for risk assessment of children with specific DNA markers could be beneficial and such markers could also help in early detection and follow up of disease process, prior to emergence of currently available disease markers such as islet autoantibodies.

10. *The results and discussion are lengthy, and not very well organized. In part this seems to be the case because there does not seem to be an overarching message. This vagueness is reflected in the abstract, which requires more specificity when describing the goal, results, and Interpretation.*

Response: Thank you for pointing this out. We have rewritten the abstract to more clearly state the specific results and to better summarize the goal of the analysis. We have also revised the text of the results section and added more subsection headings to make the flow of the analysis more easy to follow. Discussion section has been restructured to follow the structure of the results section.

11. The authors don't really introduce the importance of HAdV infections before discussing integration into the analysis

Response: Thank you for pointing this out. We have added text concerning the importance of HAdV (page 3, third paragraph) along with citations as a risk factor to T1D and IA.

12. The label of Figure 3 is not very clear and is confusing. For figure3 d,e, what does each column and row represent? These should be Labeled.

Response: Rows and columns represent the different cell types on which the correlation analysis was performed. We thank you for pinpointing the missing information and have now improved the figure legend.

13. The authors suggest that altered GSTM1 expression could be a proxy for genetic risk. This seems quite speculative. Moreover, don't they have genetic data such that this notion could be tested?

Response: We thank the reviewer for pointing this out. As responded to Reviewer #1 above, we have included genotype information from whole genome sequencing data to confirm this.

Reviewer #3 (expert in viral aetiology of endocrine disorders):

This is an important, complex and somewhat difficult-to-read contribution. Conclusions are of interest. Children prone to type-1 diabetes – when followed longitudinally for long times – show transcription profiles in peripheral blood that differ from those of non-diabetes-prone children not developing pancreatic autoantibodies. The altered transcription profiles refer especially to innate immunity and reveal that diabetes-prone children have a somewhat attenuated immune response to enterovirus infection (but not so much to adenovirus infection). The study attempts to integrate genetics, transcriptomics over prolonged times, and immune cell type alterations over time with the detection of two virus groups (enterovirus and adenovirus) at serial time points before the appearance (seroconversion) of autoantibodies to pancreatic islet antigens (insulin, GADA). The MS contains text with references and figures plus Suppl Materials with Tables and Figures, plus Reporting Summary.

Response: We thank the reviewer for the nice summary of our work. We have revised the manuscript structure with the aim to make the presentation less complex and more easy to read.

We have e.g. added subsection heading to the results sections and highlighted the key findings more clearly already in the abstract.

Title: the title is not clear to me. I may suggest: “TEDDY Study of type 1 diabetes: peripheral blood transcriptomics reveal two distinct pathogenic paths associated with the autoimmune response, virus detection, and the antiviral response”

Response: We have improved the title and added the key information about blood as a source of transcriptomics data as pointed out by the reviewer.

Abstract In the present form it does not entirely reflect the results and, especially, the main conclusions as expressed in the Discussion. It should be rewritten.

Response: We have rewritten the abstract to be more concrete in terms of results.

Samples for transcriptomics (probably frozen whole blood) and for virus detection need to be indicated. Methods say that stool is one sample (in Methods it is not mentioned how stool was collected and stored). The second sample is said to be serum (abstract and elsewhere in text). Probably, the sample is plasma OR whole blood as for transcriptomics. Please clarify. The type of sample needs to be specified all over the text (also in methods) and in Figures.

Response: Thank you for your comment. We have added these details to the methods section and improved the presentation to be more clear. Whole blood was collected with applied Biosystems Tempus blood RNA tubes and frozen. We have added details concerning the consistent TEDDY procedure for virome sample collection, storage and shipping. The second sample for virome is plasma. After data harmonizing of matched case and control samples on TEDDY due month, virome profiling consists of 9072 stool samples (mean 9 per subject) and 4686 plasma samples (mean 5 per subject) analyzed for virome. We have also simplified Figure 1a as recommended and additional collection details and virome designs from Stewart and colleagues (Stewart et al. 2018) are added to Methods (page 13-14, *Sample processing and sequencing* and *Harmonizing subject omics samples* sections).

Findings are innovative and derived from extensive studies. The main conclusion is that children genetically predisposed to type 1 diabetes seem to have an attenuated and partially ineffective antiviral response (mainly to enteroviruses) that could favor virus persistence, chronic inflammation, and the production of autoantibodies to insulin and/or GADA. The findings succeed in enlightening the recognized relation between genetic and environmental factors in the origin of type 1 diabetes. In addition, they confirm the already known pathogenic heterogeneity within type 1 diabetes. Thus this contribution may have notable implications in translational medicine.

Response: We thank the reviewer for a clear summary of key results.

Some points need clarification: Based on conversion to autoantibody production, two subgroups of type 1 diabetes have been studied: a) first insulin, b) first GADA. The groups should be indicated IA-first and GADA-first (all over the text, Tables and Figures)

Response: Thank you for the comment, we have unified the terminology throughout the manuscript and supplementary tables. We are avoiding usage of IA as abbreviation for islet autoimmunity while consistently use IAA for autoantibodies to insulin and abbreviating islet autoantibodies as IAbs. Based on the reviewer's recommendations, the two subgroups will be indicated as IAA-first and GADA-first.

Line 69: stool – as previously indicated, please insert collection and storage. Then: why stool? In fact, previous work from the Authors' group showed that virus in stool does not predict autoantibody seroconversion, nor development of clinical diabetes. Probably, the sample was available and has been studied. However, no mention of stool results appears in Results or Discussion. If no important data have been obtained from studies, I'm suggesting to delete this specimen.

Response: Thank you for this comment, we have added TEDDY collection and storage of stool samples in the methods (page 13). Our analysis with the described stool virome study confirms the T1D islet autoimmunity NCC1 enterovirus associations, previously reported by Vehik *et al.* (2021). We extend this prior analysis by incorporating plasma virome samples. In our analysis viruses identified either in stool or plasma samples are included. We have clarified this in the manuscript. Within the results, we have reported Coxsackievirus associations, belonging to the Enterovirus B species. Moreover, we have also reported and discussed host-response transcriptomic response results based on enterovirus before and after exposures (page 12, third paragraph).

Line 88: these environmental... It is not clear to me which determinants. Please rephrase.

Response: We use the term "environmental" broadly to include any external factors associated with T1D. In our analysis, we specifically focus on virus exposures as an environmental determinant. We have clarified this in the manuscript.

Line 103: data of cohorts. The study design is difficult to follow. I may suggest to insert Table 1 (Table S1 currently in Suppl Materials) with cases and controls. NCC: please define.

Response: We have improved the Figure 1 to better highlight the samples and data. Details of the cohort are included in a revised Supplementary Table S1. The NCC abbreviation (nested case-control) has been replaced with NCC1 to differentiate between the ongoing TEDDY NCC2 design.

Figure 1: rather complex to follow the numbers: 418 (1:1), then 383 Microbiome and 370 IA (full?) transcriptome. The stool, and (again) plasma (? , whole blood?), then RNAseq (is it whole blood?)

Response: As suggested, we have updated the figure with details to improve the clarity. In addition, we revised Supplementary Table 1 to ease the complexity of the overall omics sample availability of the NCC1 (initial TEDDY nested case-control cohort).

Response Table 3: TEDDY islet autoimmunity NCC1 (Nested case-control) omics data are harmonized on case control matched sample due number. Data Harmonization denotes matching on TEDDY collection identifier ‘due number’ which corresponds to due month number (incremented from initial 3-month visit). To address model sparsity, 1693 matched samples are selected in timepoints 3, 6, 9 and 12 months prior to persistent islet autoimmunity onset.

Omics	Total Samples	Islet autoimmunity NCC1 harmonized case and control children pairs	Harmonized Sample pairs in NCC1 case and control children
Stool Virome	9911	386	4536
Plasma Virome	4779	387	2343
Transcriptome (Whole Blood)	4324	370	2376

IA:islet autoimmunity, NCC1: nested case-control;

Line 145: using HLA status covariate. Unclear. Rephrase for clarity.

Response: We have revised the text to be more specific: “HLA genotype of the subject as a covariate” and define specific HLA subtypes in methods.

Figure 2: Full IA TAS2R30 not considered. GADA first, only GSTM1 considered. Why? Please comment at least in the legend.

Response: We have selected representative genes from each islet autoimmunity group as an example. Selection criteria was known biological function of the genes to enable us to discuss their putative role in T1D.

Discussion is well organized. Since conclusions are numerous, I would suggest to number them 1 to 8: abnormalities of the immune system; two different pathways; results consistent with the association of prolonged or cumulative EV infection with IA production; defects of antiviral responses more relevant to EV compared to HAdV; confirmed relevance of GSTM1 to type 1 diabetes; CVB apparently more relevant to autoimmunity than other EV genotypes; possible role of eosinophil profile; increase of monocytes and decreased B cell proportions.

Response: According to journal guidelines, we are not allowed to do explicit numbering. We have tried to further improve the structure of the discussion section by reorganizing the text to follow the (revised) structure of the results section.

Line 425: Clarify whole blood (Na-EDTA or K-EDTA)? Then, detection of adenoviruses: was the assay based on RNA (virus transcripts), or DNA (virus genome)? If DNA, please add how it was extracted and prepared for NGS:

Response: For RNA sequencing, whole blood samples were collected with Applied Biosystems Tempus blood RNA tubes. Plasma and stool samples were used for virome analysis. We have added more details about processing to the methods section. Adenoviruses are based on DNA (virus genome) and the extraction/preparation have been previously reported, that we cited, by TEDDY projects Stewart *et al.* and Vehik *et al.* As previously reported (Lee *et al.* Diabetologia. 2013 Aug;56(8):1705-1711. Plasma virome is processed from Total nucleic acids extracted from 250 µl plasma (NucliSENS easyMag, Biomerieux, France) and nucleic acid quantity and quality assessed (2100 Bioanalyzer, Agilent Technologies, USA). As reported previously in Vehik *et al.* filtrates of stools (~0.15–0.20 mg dispersed in 100–200 µl saline and passed through a 1.2-µm filter) were directly extracted for total nucleic acids, then incubated on mixtures of four cell lines (Hela cells, Vero cells, RD cells expressing coxsackievirus and adenovirus receptor, and HEK-293 cells; 25% each, plated at 40% overall confluency) in Dulbecco's modified Eagle medium containing 2% calf serum for 6 d, to amplify the viruses present at very low levels. Cell lines were chosen for the breadth of virus replicative efficiency of type A, B and C enteroviruses, as well as other common viruses. Infected cultures were not passaged. Cells and supernatants were collected for total nucleic acid extraction and analysis.

Line 450: "serotyping": it is not serotyping, it is genotyping. Change all over the text.

Response: Thank you for pointing this out. We have changed the terminology to genotyping throughout.

References: adequate.

Legend to Figures (main and supplementary): adequate

Reporting Summary: OK

Response: Thank you for all the valuable comments.

REVIEWERS' COMMENTS

Reviewer #1 (Remarks to the Author):

The Authors have answered in an adequate way my concerns and the present version of the manuscript is clearly improved, providing relevant information for the field.

Minor points:

Page 6, line 205, the word "data" appears twice

On page 10, line 337, remove "though importantly".

Reviewer #2 (Remarks to the Author):

The authors have been responsive to critiques from the reviewers. Overall, the message and data presentation in this revised version of the manuscript have been improved. In particular, the authors have done a nice job of highlighting key take away messages. The added genetic and proteomic data are a strength.

Reviewer #3 (Remarks to the Author):

The revised MS has greatly improved upon resolving methodological issues and Reviewers' criticisms. The abstract, text and discussion have been re-written. Conclusions are now supported by more clear Figures and Tables. This Reviewer's indications have been followed adequately. Results demonstrate immune related blood cell transcriptomic differences between cases and control children prior to islet autoimmunity. It is also well documented that enteroviral infection induces less robust antiviral response in children who later develop islet autoimmunity as compared to control children. The role of the GSMT1 gene in susceptibility to type 1 diabetes is suggested by both transcriptomics and genomics. The issue is now open to further investigation.

REVIEWERS' COMMENTS

Reviewer #1 (Remarks to the Author):

The Authors have answered in an adequate way my concerns and the present version of the manuscript is clearly improved, providing relevant information for the field.

Minor points:

Page 6, line 205, the word "data" appears twice

On page 10, line 337, remove "though importantly".

RESPONSE: We thank the reviewer for all valuable comments to improve our work. We have implemented the minor points raised in the final version.

Reviewer #2 (Remarks to the Author):

The authors have been responsive to critiques from the reviewers. Overall, the message and data presentation in this revised version of the manuscript have been improved. In particular, the authors have done a nice job of highlighting key take away messages. The added genetic and proteomic data are a strength.

RESPONSE: We thank the reviewer for all valuable comments to improve our work through the review process.

Reviewer #3 (Remarks to the Author):

The revised MS has greatly improved upon resolving methodological issues and Reviewers' criticisms. The abstract, text and discussion have been re-written. Conclusions are now supported by more clear Figures and Tables. This Reviewer's indications have been followed adequately. Results demonstrate immune related blood cell transcriptomic differences between cases and control children prior to islet autoimmunity. It is also well documented that enteroviral infection induces less robust antiviral response in children who later develop islet autoimmunity as compared to control children. The role of the GSMT1 gene in susceptibility to type 1 diabetes is suggested by both transcriptomics and genomics. The issue is now open to further investigation.

RESPONSE: We thank the reviewer for all valuable comments to improve our work through the review process.